



**Evolution of Vegetation System in Heihe River Basin in the last 2000 years**
Shoubo Li[1], Yan Zhao[2, *], Yongping Wei[2], Hang Zheng[3]
* Corresponding author
1 School of Geography and Remote Sensing, Nanjing University of Information Science and
Technology, Nanjing, China 210044
2 School of Geography, Planning and Environmental Management, the University of Queensland,
Brisbane, Australia 4072
3 State Key Lab of Hydroscience and Engineering, Department of Hydraulic Engineering,
Tsinghua University, Beijing, China 100084
Abstract: The response of vegetation system to the long-term changes in climate, hydrology, and
social-economy in a river basin is critical for sustainable river basin management. This study aims
to investigate the evolution of natural and crop vegetation systems in Heihe River Basin (HRB)
over the past 2000 years. Archived Landsat images were applied to derive vegetation spatial extent
and biomass for 1987 to 2015.The area and biomass of the vegetation before 1987 were
reconstructed based on previous research results the derived relationship between the vegetation
biomass and climatic and hydrological variables in the last 30 years with instrumental data. The
key findings are: 1) both natural and crop vegetation have gone three development stages:
Pre-development stage (before 1949), rapid development stage (1949-2000), and
post-development stage (after 2000); 2) there was a much faster increase of crop biomass than that
of native vegetation since 1949, and 3) the ratio of natural vegetation to crop vegetation decreased
from 16 at Yuan Dynasty to at about 2.2 since 2005. This ratio represents the land and water
development at river basin at changing climate and social-economy, it could be used as an
indicator to plan the objective or examine the outcome of water and land management at river
basin.

Key Words: Natural vegetation, crop vegetation, biomass, remote sensing, reconstruction, river
basin





## 1. Introduction

Natural vegetation plays a key role in maintaining functions of catchment ecosystems including contributions to goods, services, and ecosystem biodiversity in arid and semiarid river basins (Ahlström et al., 2015; Feng et al., 2013; Kefi et al., 2007). With the rapid growth of population, increasing amount of water worldwide has been allocated to support human activities, particularly for irrigation, whereas water for natural vegetation, wetland, and other catchment ecosystems might have been compromised. Consequently, natural vegetation systems in water-limited regions have been degraded, and salinization and desertification have been reported repeatedly (Huang et al., 2015; Li et al., 2007; Su et al., 2007; Xue et al., 2015). To understand the development of natural vegetation under different water conditions and its interactions with the crop system is vital for the sustainable river basin management.

There are overwhelming studies on the impact of land use and land cover changes driven by either human activities or climate changes on the catchment hydrological regime and the water cycle (Esteban and Albiac, 2011; Ian and Reed, 2012; Leggett et al., 2003; Xue et al., 2015). However, few studies have been found to investigate how the vegetation system evolved to accommodate the changes in water regimes at the basin scale. In the last decade, an increasing number of studies have contributed to the knowledge of allocating the limited water resources among different ecosystems in order to balance the economic development and environmental sustainability (Wang et al, 2007). However, most of these studies were carried out at a short time scale, either to identify the rationality of water allocation schemes reform (Cheng, 2002; Yang et al, 2003) or to test the effectiveness of ecological restoration projects (Thevs et al 2015). Long-term change in vegetation system in response to significant alternations in climate, hydrology, and social-economy is missed in current literature (Sivapalan et al, 2012).

The knowledge gap identified above happened partly due to the unavailability of long-term instrumental data on vegetation and hydrological change at the basin scale. With the rapid development of remote sensing technique, images acquired from multiple satellite platforms provides an ideal method to track the landscape changes in river basins in the past five decades (Beuchle et al, 2015). Among a mass of the remote sensing based metrics to characterize vegetation system, spatial extent or area, normalized differential vegetation index (*NDVI*) and biomass are commonly recognized as the most effective indices to reflect the status of the vegetation and widely applied in spatial analysis of landscape ecosystems (Pettorellia et al, 2005; Pinsky and Fogarty, 2012). For the historical periods earlier than five decades from now, emerging approaches including dendrochronology, ice core analysis, and other empirical methods have enabled the possibilities of reconstructions of eco-hydrological elements and their long-term variations (Turner et al., 2007; Lowry and Morrill, 2011). However, few attentions have been paid to historical landscapes and most of the limited existing reconstructions focused on the cultivated area in historical records (Xie et al, 2013; Ramankutty and Foley, 1999).

The Heihe River Basin (HRB), located in arid North-western China, is an important part of the ancient Silk Road established in the Han Dynasty (206 BC - AD 220). It was also a trade center between China and western countries, which facilitated a cultural and economic exchange for approximately 1500 years. HRB is a typical inland river ecosystem, which includes natural





vegetation, irrigated crop, desert and terminal lakes. Increasing agricultural development and changing climate and hydrology over the past 2000 years have significantly changed the way of land and water resources use and modified the catchment vegetation system (Lu et al, 2015, Yan et al, 2016). Therefore, HRB is an ideal study area for investigating the evolution of vegetation system at river basin for a long time frame.

This paper aims to understand the evolution of vegetation system in HRB over the past 2000 years in which natural vegetation and crop vegetation were considered. Specifically, it includes three objectives: 1) to determine the area and biomass of vegetation using remote sensing imagery for recent years (since 1987); 2) to reconstruct vegetation distribution and biomass levels for previous periods (before 1987) and 3) to determine potential driving factors for vegetation developments. It is expected that the methods developed and the findings obtained from this study could assist to understand how current ecosystem problems were created in the past and what are their implications for future river basin management.

## 2.    Material and Methods

### 2.1. Study area

HRB is the second largest inland river basin of China, which stretches between 38 °- 43 °N and 98 ° - 102 °E (Figure 1). The middle and lower course of HRB are occupied with different landscapes including river delta plain, terminal lakes, moving and semi moving dunes, and low mountains and hills. The unused land such as Gobi desert and bare land accounts for more than 75% of the river basin while cropland only takes up 4%. The rest of the landscape is natural oasis in which the main vegetation types are dry steppes and shelter forests.





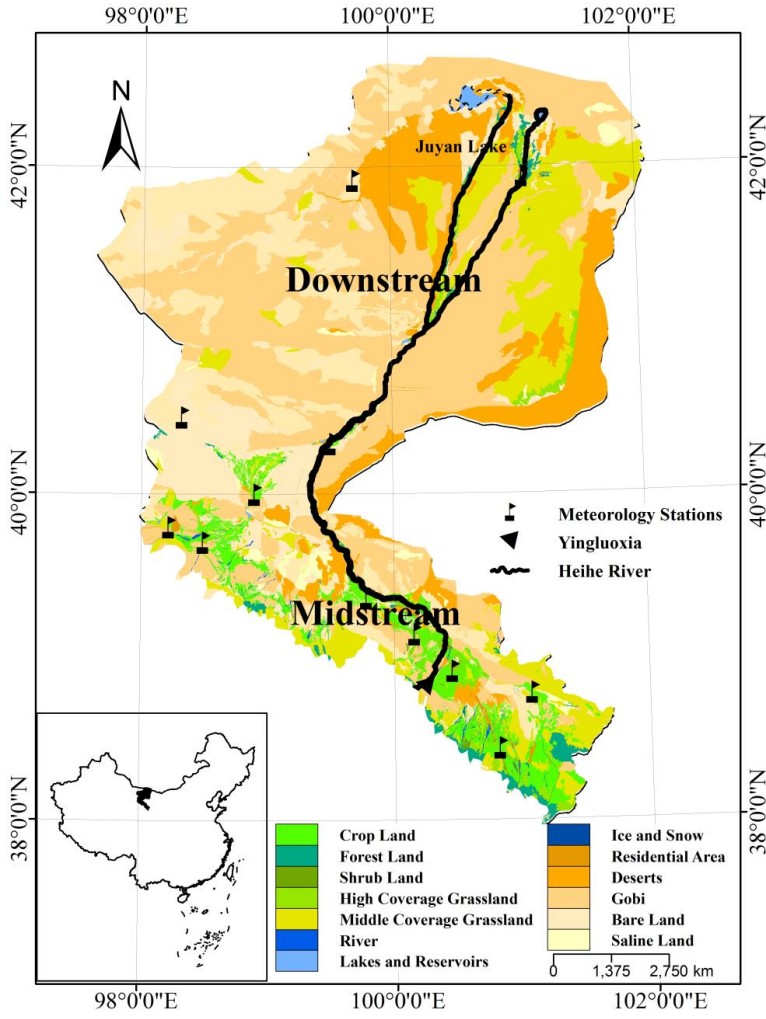

Figure 1: Location of the Heihe River Basin (HRB). Land cover data (2011) available at WestDC
database (http://westdc.westgis.ac.cn/)
The terrain in HRB is a gradual tilt from southwest to northeast. The altitude of the area ranged
from 820 to 1100 m above sea level. The region is occupied with a typical continental arid climate
characterized by frequent wind, scarce rainfall, abundant sunshine and high evaporation. Annual
average temperature in this area is 8.3 °C during the last three decades with remarkable seasonal
variations. Temperature could decrease to -37.6 °C in winter months while highest temperature
normally happened in July, which could fetch up to 43.1 °C. The annual average pan evaporation
in the Ejina oasis is 3,749 to 4,132 mm/year, which is much higher than mean annual precipitation
(ranged from 7 to 101 mm/year) with substantial interannual variations over the past three
decades.
Benefiting from the Heihe River originated in the North of Qilian Mountains of Tibetan Plateau,
HRB has experienced intensive agricultural activities to meet the grain demands of military events



since Han Dynasty (121 BC – 220 AD) (Xie et al., 2013). Nowadays, the midstream is still one of the most important agricultural belts in Northwest China. However, the increasing water abstraction for irrigating, along with elevated usage for domestic purposes in middle reaches, has substantially consumed the water for downstream systems over the past several decades. Consequently, the Juyan Lake shrank dramatically in last 100 years, and dried-up in 1992. Since the late 1990s, the Chinese government implemented series of policies to ensure that water delivered to lower course of the basin was enough to sustain the ecosystems and avoid any further degradations. In 2002, the Juyan Lake started to retain water again which was taken as an important sign of ecosystem recovery.

**2.2. Study period**

We selected the past 2000 years as our study period, which started from the Han Dynasty (206 BC – AD 220) (Table 1). This timescale covered several ancient dynasties of China, the Republic of China (RC), and the Peoples' Republic of China. The period has experienced dramatic changes in climate, land use, runoff, management policy, population, social and ecological developments. All these factors could contribute to changes in water cycles within the river basin and, therefore, influence vegetation distributions.

Table 1: major dynasties during the selected study period (Lu, et al., 2015).

| Dynasty | Period | Main production |
|---|---|---|
| Han Dynasty | 206 BC – AD 220 | Agriculture |
| Wei-Jin Era | AD 220 – AD 420 | Animal husbandry |
| Tang Dynasty | AD 618 – AD 907 | Agriculture |
| Yuan Dynasty | AD 1271 – AD 1368 | Animal husbandry |
| Ming Dynasty | AD 1368 – AD1644 | Agriculture |
| Qing Dynasty | AD 1644 – AD 1912 | Agriculture |
| Republic of China (RC) | AD 1912 – AD 1949 | Agriculture |
| The Peoples' Republic of China (PRC) | Since AD 1949 | Agriculture |

**2.3. Determining vegetation distribution and estimating vegetation biomass**

**2.3.1.  Landsat image preprocessing**

We used all available cloud-free Landsat images in HRB to derive vegetation dynamics for 1987 to 2015. Five Landsat scenes (path/row of 133/31, 133/32,133/33,134/31 and 134/32) for each year were required to cover the area. The collected images covered the timescale ranging from 1987 to 2015 except for 1989 and 1996 when there were no high-quality images. Most of these images were acquired during late summer and early autumn (from June to October) to represent the growing season for crops and natural vegetation in the study area.

The data products containing digital numbers (DN) were downloaded from the United States Geological Survey (USGS) Earthexplorer website (http://earthexplorer.usgs.gov/). The DN values were converted into top-of-atmosphere (TOA) reflectance using the radiometric gain and offset values associated with each Landsat images. Then a Quick Atmosphere Correction (QUAC) method was adopted to account for atmospheric scattering and to derive land surface reflectance in



order to ensure that the change detection analysis truly detected changes at the Earth's surface
rather than solar illumination differences or potential differences in atmospheric conditions. The
normalized Differential Vegetation Index (*NDVI*) was then calculated using the red and
near-infrared bands for each year.
As demonstrated in Figure 2 with phenology profile of natural and crop vegetation derived from
Moderate Resolution Imaging Spectroradiometer. (MODIS), *NDVI* presented significant seasonal
variations. Since the collected Landsat images were acquired at different dates (sometimes
different months) of a year, the above calculated *NDVI* values would have included this seasonal
variations and not suitable for inter annual comparisons. To compensate this effect, we used the
MODIS *NDVI* profile in 2013 to calibrate the Landsat *NDVIs* to annual maximum *NDVI*, which
could effectively reflect the same growth stage of vegetation in multiple years, using a linear
interpolation algorithm. Specifically, with the knowledge of acquisition date of Landsat image for
a specific year, the ratio of MODIS *NDVI* for that date to the maximum MODIS *NDVI* was
calculated and this ratio was applied to Landsat derived *NDVI* to estimate maximum *NDVI* for that
year.

$$NDVI_{L\text{-}max} = NDVI_{Li} \times NDVI_{M\text{-}max} / NDVI_{Mi}$$

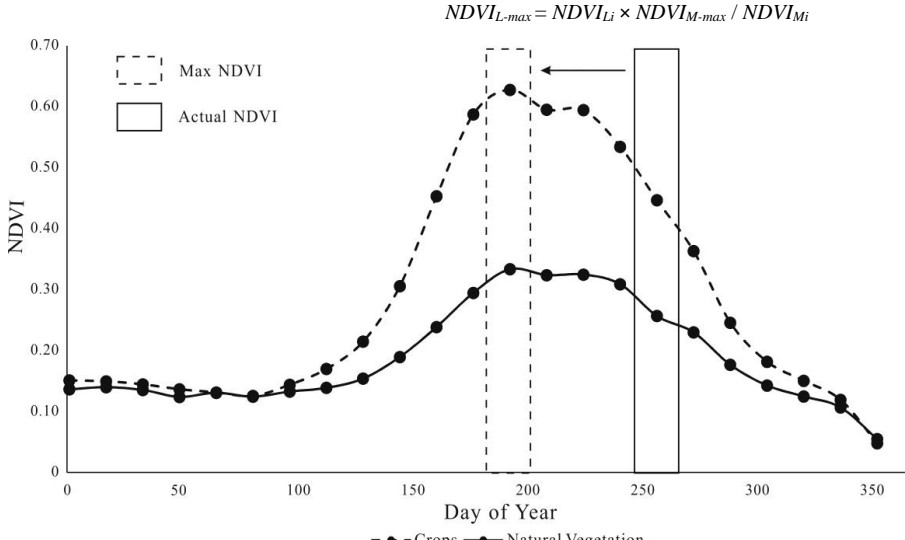

Figure 2: *NDVI* profile for natural vegetation and crops in 2013. The equation in the figure indicated
the scheme for calibrating the actual *NDVI* values (L stands for Landsat, M stands for MODIS, i stands
for the date when actual *NDVI* need calibration)

**2.3.2    Determining vegetation areas and biomass since the 1980s with satellite images**
Two rounds of threshold analysis were applied to determine natural and crop vegetation
distributions in HRB. Non-vegetation landscapes, including water surfaces, deserts, residential
areas and other bare surfaces, were masked out through analyzing the *NDVI* histogram distribution
characteristics, selecting 0.12 as the first threshold value and making minor verify for each year.





Then, the second round of threshold analysis was introduced to separate crop vegetation from natural vegetation according to their phenology differences (Figure 2). Briefly, we randomly sampled the vegetation maps and derived the average *NDVI* levels for natural and crop vegetation, respectively. A most recent land cover map (2011) created by the Cold and Arid Regions Environmental and Engineering Research Institute, Chinese Academy of Sciences was introduced to assist identifying vegetation types. The preliminary result was first overlaid with each year's images to check the accuracy. In addition, the results in 2000 and 2011 were verified with a set of land use maps (2000, 2011) obtained from the WestDC database (http://westdc.westgis.ac.cn/). A detailed scheme of inter-comparison of land cover maps between this study and existing results were detailed in Zhao et al., (2016). Overall, the two datasets presented substantial consistency where kappa coefficients (k) were 0.7206 and 0.6731 for 2000 and 2011. Areas of natural and crop vegetation for each year were calculated by summing areas of every small patch of natural and crop vegetation respectively.

In order to study the vegetation development and water usage, we calculated biomass based on NDVI. Regression models established by Zhao et al. (2006, 2010) were adopted to quantify biomass for both natural and crop vegetation. In those research, the herbaceous biomass (g) was measured by dry biological weighing methods and the field measured biomass for natural and crop vegetation in this region showed high correlation with *NDVI* ($R > 85\%$ and $p < 0.01$) in the same area and the following formula was established for natural and crop vegetation respectively.

$$\text{Biomass} = 327.4 \times \text{NDVI} + 102.29 \quad \text{for natural vegetation} \quad \text{(Zhao et al. 2006) (1)}$$

$$\text{Biomass} = 1,789 \times \text{NDVI} + 559.68 \quad \text{for crops} \quad \text{(Zhao et al. 2010) (2)}$$

We used the equations above to calculate the biomass from 1980 pixel by pixel each year and use regional statistics method to acquire natural vegetation biomass and crops biomass of the study area.

### 2.3.3 Reconstructing vegetation distribution and biomass levels in historical periods

Lu (2015) reconstructed vegetation distribution in past 2000 year in our study area to analyze the evolution of human–water relationships. In Lu's study, the historical cropland was reconstructed based on population, grain yield and ancient ruins distributions (Xie, 2013). The area of natural vegetation was estimated based on two assumptions: (1) people selected the regions with natural oases (grassland and forest) rather than desert for reclamation in the historical periods because the former have better water and soil conditions in these arid regions, and (2) once the reclaimed farmlands were abandoned and no vegetation was covered, they were subsequently decertified (Lu et al., 2015). The crop and natural vegetation area of each dynasty was calculated based on Lu's results.

The vegetation biomass in the historical periods was not available in the literature. Using the satellite-based results since the 1980s, we reconstructed the vegetation biomass based on its relationship with several variables which could have impacts on the vegetation development. The selected variables are temperature ($T$), river flow from upstream ($Q$), river flow to Juyan Lake ($Q_l$), groundwater recharge ($Q_g$) and precipitation ($P$). $T$ and $P$ records were collected from the surrounding weather station (Figure 1). River flow to Juyan Lake was determined according to the records measured at Ejina station. Streamflow through Yingluoxia gauge station stands for the



total upstream inflow to the study area ($Q$). Groundwater reserves data were obtained from the
government statistics yearbook. The component of streamflow consumed for vegetation
developments ($\Delta Q$) was then determined by deducting $Q_l$ and $Q_g$ from $Q$. With this established
database, a stepwise regression method was introduced to explore relationships between biomass
(biomass for natural vegetation, biomass for crops and, the total biomass) and the selected
hydrological and climatic metrics. The significant correlation among total biomass (g), $\Delta Q$ and $T$
were found ($R^2 = 0.612$). As indicated by the regression model, both temperature ($T$) and water
supply ($\Delta Q$) present positive effects over vegetation productivity.
$Total\ Biomass(g) = 39.246 * T + 9.312 * \Delta Q - 345.671$   (3)
$\Delta Q = Q - Q_l - Q_g$      (4)
We then applied equation 3 to estimate historical total biomass levels using the corresponding ($T$,
$\Delta Q$) settings. Specifically, historical $T$ was derived from paleoclimate records reported by Yang
(2002). $Q$ estimations by Sakai et al (2012) based on glacier mass balance analysis were adopted.
Since the spatial extent of the lake did not change much in historical periods, $Q_l$ was assumed to
be equal to evaporations from the lake surface which could be derived from public articles (Xiao
and Xiao, 2008).; $Q_g$ was set to be 0 based on the assumption that groundwater level did not
change over the historic periods when agricultural activities were relatively small.
Unlike crop biomass which was largely influenced by technologic development, the biomass of
natural vegetation was much less influenced by human activities. Therefore, we made an
assumption that biomass density for natural vegetation over the study period did not change.
According to the Landsat-based results for the past 30 years, the average biomass density for
natural vegetation was stable at about 190 g/m$^2$ and this value was applied to historical vegetation
maps to get the corresponding biomass estimations for natural vegetation. Crop biomass was then
estimated by deducting the component for natural vegetation from the total biomass estimations.
Biomasses for historical natural and crop vegetation were further estimated.
**2.4  Determination of potential driving factors for vegetation developments**
Multiple linear regression analysis was adopted to investigate the potential driving factors causing
changes in spatial extent of vegetation and its biomass levels. Hydrological variables including $Q$,
$\Delta Q$ and climatic variables including $P$ and $T$ were related to spatial extents and biomass levels of
natural and crop vegetation to find if there were any significant relationships. These selected
variables were also taken as independent variables to find the quantitative models between these
variables and the vegetation spatial extents. As the reconstruction data for the historical periods
might incur great uncertainties in records and estimations, the regression analysis was conducted
for the whole study period and the recent decades with instrumental data respectively. The analysis
was performed with the IBM SPSS Statistics software package (version 20.0).
**3   Results**
**3.1 Spatial and temporal variations in vegetation distribution in the past 2000 years**
The reconstructed natural (green) and crop (red) vegetation distributions in the past 2000 years are
shown in Figure 3. Historic maps (before 1987) were derived from Lu's results (Lu et al., 2015)



and maps after that were interpreted from Landsat images. The spatial extent of crop vegetation in
both midstream and downstream of HRB has changed significantly over the study period. Historic
distribution of crops was focused in relatively small regions with certain variations. It was until
the establishment of the PRC the crop vegetation started to increase at a high rate. As clearly
demonstrated in the maps, there were crop areas distributed in downstream regions around Juyan
Lakes in historic periods, however, in the modern China, crops were distributed mainly in the
middle basin of HRB. As for natural vegetation, there were few changes in the midstream regions.
From 1949 to 2000, natural vegetation in the midstream basin has substantially increased with
large inter-annual fluctuations. After 2000, vegetation distribution was relatively stable at a high
level. The downstream vegetation has experienced gradually increase corresponding to the crop
area decreasing during these periods.







Figure 3: Reconstructed natural (green) and crop (red) vegetation distributions in the past 2000 years.
Historic maps (before 1987) were derived from Lu's results (Lu et al., 2015) and maps after that were
interpreted from Landsat images.
The temporal variation of vegetation areas over the past 2000 years is presented in Figure 4. The





total vegetation area increased by 8,732 km$^2$ during the studied period. Historically, total
vegetation within HRB experienced a slight decrease, from about 8,122 km$^2$ in Han Dynasty to
about 6,918 km$^2$ in the Republic of China. Natural vegetation for this period constantly decreased
by 21% to only 5,000 km$^2$. Cropland for the same period presented more variations: It had a
large spatial extent at Han Dynasty at about 1,755 km$^2$, and then gradually decreased to about 379
km$^2$ in Yuan Dynasty. From the Ming Dynasty, it started to increase again and reached a peak of
1,917 km$^2$ in the Republic of China. Situations were different in the period of modern China. Total
vegetation area increased from about 6,918 km$^2$ to 11,362 km$^2$ in 1987 and to 13,863 km$^2$ in 2000
with an increasing rate of 2% per year, while the crops have substantially increased by about 150%
to 4,939 km$^2$ in 2000. In the same period, natural vegetation has also substantially increased from
about 6,559 in 1987 to about 8,924 km$^2$ in 2000. After 2000, the increasing rate of the crop has
decreased from 3% per year to 0.3% per year while the natural vegetation has substantially
increased to about 11,691 km$^2$ in 2013, resulting in the total vegetation area keeping increase
steadily to 16,854 km$^2$ in 2013.

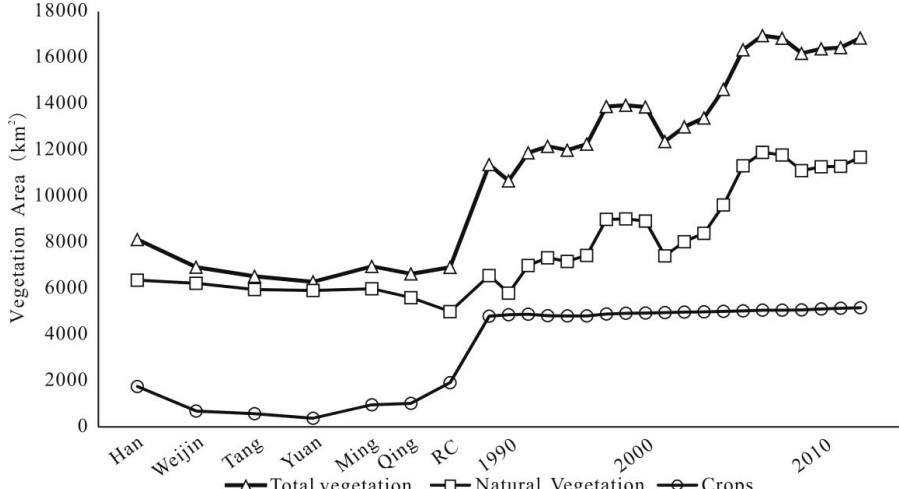

Figure 4: Temporal variations in total vegetation areas (triangle), natural vegetation (square) and crops
(circle). It should be noted that data after 1987 was the results after applying a 3 years moving average
to reduce the annual fluctuations.

The ratio of natural vegetation area to crop vegetation areas varied over the past 2000 years
(Figure 5). The ratio, to some extent, reflected the relationship and interactions between the two
vegetation systems. As demonstrated in Figure 5, although small in scale, natural vegetation
occupied a major portion of the vegetation in this area in Han Dynasty and it substantially
increased until Ming Dynasty when the ratio peaked at 16. The increased ratio during this period
could be attributed to the degraded farming activities (Figure 3, Figure 4). As agriculture started to
boom since Qing Dynasty, the ratio decreased significantly to about 1.4 in the Republic of China.
Afterward, the ratio showed a constant increase with a rate of 0.06 per year ($R^2 = 0.8063$). Overall,
the ratio natural vegetation area to crop vegetation areas during the modern China was relative
stable compared with the great historic fluctuations and it is stabilized at around 2.2 since 2005.



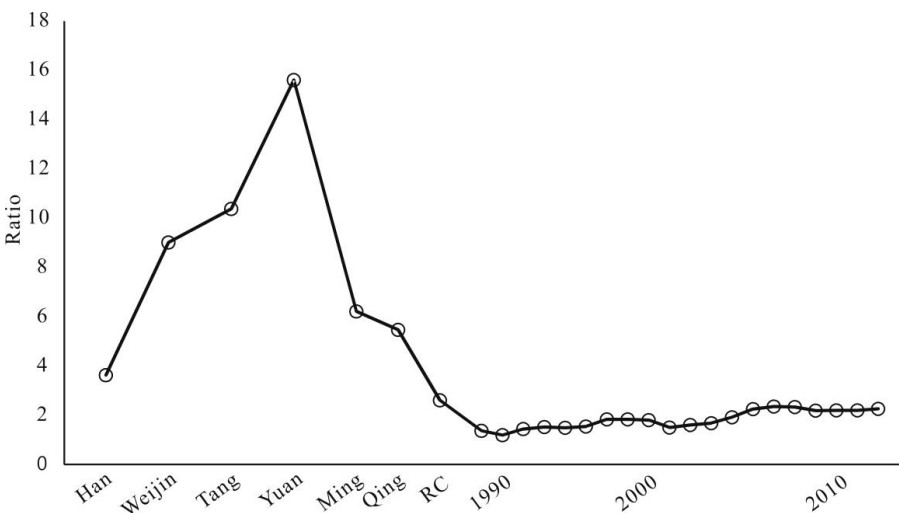

Figure 5: Changes in ratio between the areas of natural vegetation to crop vegetation in the past 2000
years

## 3.2  Changes in vegetation biomass over the past 2000 years.

Using the Landsat-derived biomass estimations and the corresponding hydrological and climatic
records, we produced the relationship between biomass and T and streamflow as demonstrated in
equation 3. The relationship was applied to the long term T and streamflow records to derive
historic biomass estimations. As showed in Figure 6, biomass in natural vegetation in historic
periods (Han Dynasty to the Republic of China) had experienced a slight decrease by 20% and the
biomass of crop decrease before Tang Dynasty and increased after. Since the Republic of China,
biomass in natural vegetation has shown gradually increase from about $95 * 10^4$ t to $159 * 10^4$ t in
2000. After 2000, the upward trend continued with a higher increasing rate was observed. For
crops, the annual biomass presented a sharp increase trend since 1949 by about 4 time and slight
increase trend in past 30 years. The average productivity per unit area of natural vegetation was
stable while the average productivity per unit area of crop increased by 2.2 times in past 2000
years. The average productive per unit area of crop increased by about 180% since PRC.





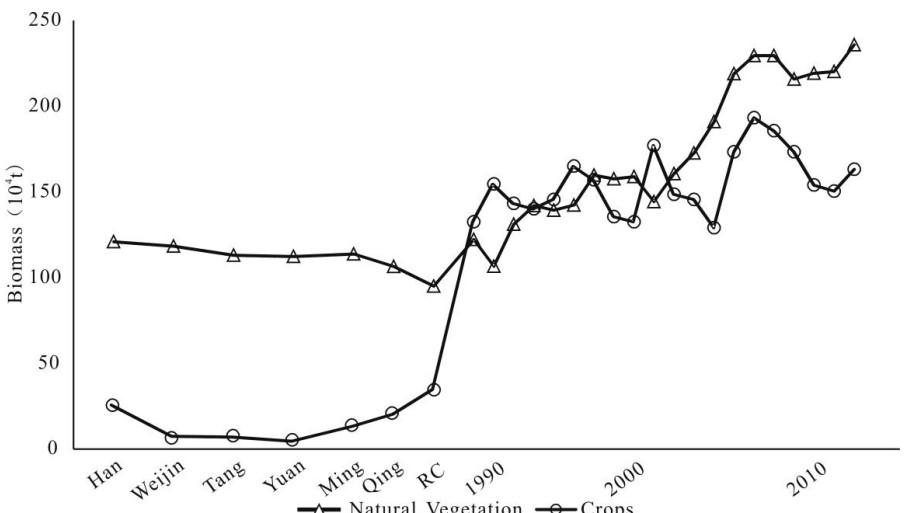

Figure 6: Temporal variations in biomass of natural vegetation (triangle) and crops (circle) over the
past 2000 years
3.3 **Impacts of hydrological and climatic variables over vegetation development in the past**
**2000 years**
The regression analysis on the relationship between hydrological and climatic variables and
vegetation development show that both $T$ and $\Delta Q$ presented an overall positive effect on natural
and crop vegetation distributions (Figure 7 a and b). From a holistic perspective, $T$ showed a
significant impact over natural vegetation expansion while its effects on crops were quite limited.
Meanwhile, $\Delta Q$ exerted similar effects on both natural ($R^2 = 0.6016$, $p = 0.00$) and crop ($R^2 =$
0.9188, $p = 0.00$) vegetation development over the past 2000 years. It is also found that $T$ showed
significant positive impacts over both natural ($R^2 = 0.7684$, $p = 0.00$) and crop ($R^2 = 0.6836$, $p =$
0.00) during the past three decades with instrumental data (Figure7 c and d). Similar for $\Delta Q$, it
alone contributed about 77% and 60% of the area expansion since 1980s for natural ($p = 0.00$) and
crop ($p = 0.00$) vegetation, respectively. A multiple factor regression analysis shows that
increasing $T$ and $\Delta Q$ could explain over 90% of the vegetation development or 96.0% for natural
vegetation and 91.7% for crops. Although the development of vegetation did not show an obvious
relationship with precipitation, there were few years that vegetation area was less than other years
in last 30 years, for example, 1992 and 2001, may due to the inter annual variations of
precipitation.



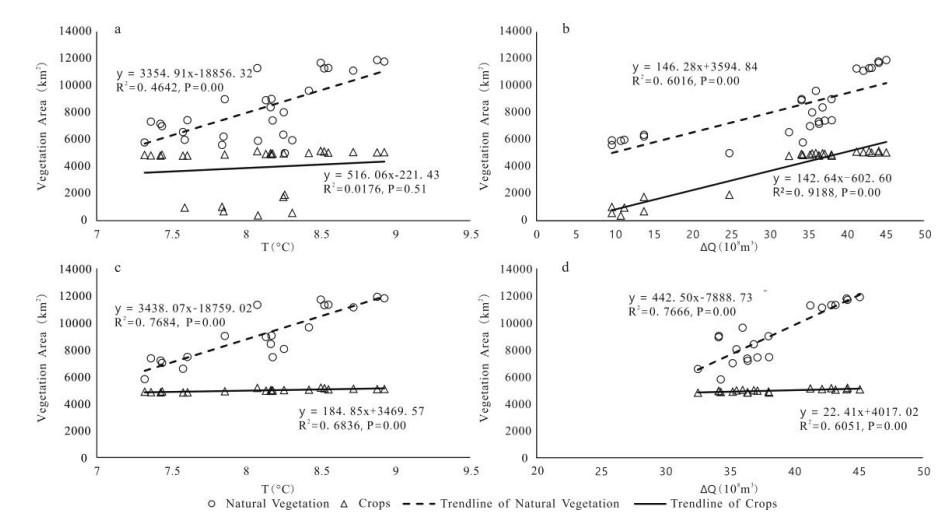

Figure 7: Correlation between vegetation (circle: natural vegetation, triangle: crops) and *T* (a, c) and
*ΔQ* (b, d). (a) and (b) presented all reconstructed data for the past 2000 years, (c) and (d) only used
Landsat-derived estimations.
## 4  Discussions and Conclusions
This study presented an empirical study of investigating the evolution of vegetation system in the
HRB over the past 2000 years. The vegetation system was categorized into natural vegetation and
crop vegetation. The area and biomass of each vegetation system since the 1980s were estimated
based on the remote sensing image data. For the historical periods, the area and biomass of each
vegetation system were reconstructed based on the relationship between the area and biomass of
the vegetation system and the climatic and hydrological variables in the last 30 years with the
measured data. Some major research findings and their implications for future research and river
basin management practice are discussed as follows:
Both natural and crop vegetation development in Heihe River Basin, based on the change in area
and biomass in the past 2000 years (Figures 3, 4, 7), can be divided into 3 stages: (1)
Pre-development stage (before 1949), (2) Rapid development stage (1949-2000), and (3)
Post-development stage (after 2000). In pre-development stage, agriculture was developed at a
lower level only for the political and military needs (Xie et al, 2013). The natural vegetation/crop
ratio came to the highest point in the Yuan Dynasty in Fig.5. During this period, the temperature
fluctuated marginally and the terminal lake area did not change much (Yang, et al., 2002; Lu, et al.,
2015). The natural vegetation showed a slight decrease trend as the runoff from upstream
decreased from $14.2 \times 10^8$ m$^3$ to $13.5 \times 10^8$ m$^3$. After 1949, crop area in HRB experienced rapid
increase as food security had been the priority agricultural policy in China. The government
encouraged farmers to reclaim unused land and promoted lots of irrigation projects (Zhang, et al.,
2015). In addition, the shelter forest system established after the 1980s not only protected the
existing cropland but also made it possible to change the desert surrounding oasis to farm. Natural



vegetation during this period also experienced rapid increase, temperature increase by 0.5 - 1°C
and upper stream runoff increase from $13.5 \times 10^8$ m³ to $15 \times 10^8$ m³ could explain it. And water
leakage from crop irrigation may also contribute to natural vegetation development. As a
consequence of the rapid development of agriculture, the terminal lake (Juyan Lake) of about 900
km² was dried up and groundwater was over-pumped for irrigation. After 2000, crop vegetation
kept relatively stable as a result of the implementation of the policy "ensure water supply to the
lower course of the basin to avoid ecosystem degradation". Natural vegetation keeps increasing
during this period because the temperature and runoff continue to increase. These stage
developments were the result of changes in agricultural and water policies and changes in climatic
and hydrologic variables.
There was a much faster increase of crop biomass than that of native vegetation since 1949
(Figure 6). The average biomass of crop per unit area increased by 180% and the biomass of
natural vegetation did not change much. Lu et al (2015) also found the agricultural water
productivity increased by 6 times in past 50 years in the middle reach of Heihe River. This is the
result of technologic progress on agriculture and water application. After 1949, especially after
reform and opening of the national economy in the late 1980s, there were great improvements in
irrigation, crop varieties, chemical fertilizers and pesticides, and mechanization in HRB.
Technological improvement influences the relationship between crop and natural vegetation.
Advances in agricultural and water technologies enabled more crop biomass without the
increase of crop area and facilitated the transfer of water from agriculture to downstream
ecological purposes without compromise of the middle stream economic benefit.
The total vegetation area in HRB has been increased by 8,732 km² in the past 2000 years. Crop
and natural vegetation presented different evolutionary pattern (Figure 3 and Figure 4) and the
ratio of natural vegetation to crop vegetation from 16 at Yuan Dynasty to at about 2.2 since 2005
(Figure 5). It is the result of the increase in human water demand from agriculture and urban
development, increase in agricultural and water technological development for improving crop
biomass, increase in water allocation for the environment (terminal lake) and increases in
temperature and upstream runoff. Any changes in these factors will bring about the change of the
ratio natural vegetation to crop vegetation. This ratio represents the land and water development at
river basin at changing climate and social-economy. Thus, it could be used as an indicator to plan
the objective or examine the outcome of water and land management at river basin. More research
is needed in future to develop an understanding of the mechanism of dynamic interaction between
natural vegetation and crop vegetation. With the knowledge of this interaction, water and land
would be better managed for the better balance between the human and natural systems in river
basins.
**Acknowledgement:**
This project is supported by the National Natural Science Foundation of China (No: 41301036,
41501464, 91625103) and the Australian Research Council (Project No: FT130100274).

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
