# Peer review of "Evolution of Vegetation System in Heihe River Basin in the last 2000 years"

_Hydrology and Earth System Sciences, 2016_

## Referee Comment (RC1) · Anonymous Referee #1 · 26 Mar 2017

General comments:

This study aims to understand the evolution of vegetation system in Heihe River Basin over the past 2000 years, by reconstructing the spatio-temporal distribution of natural vegetation and crop vegetation from those previous study results and Landsat images. Because of the challenging problems in the human-environment interaction in this region, long-term change in vegetation system in the study area should be much appreciated. This research has made some advancement in the evolution of vegetation system with long-term scale, by synthesizing remote sensing data and results from previous researches. It will promote the understanding of the ecosystem health in the region and how to achieve a sustainable management in the Heihe River Basin. Overall, this is a solid and well documented contribution that can be accepted after some moderate revisions.

[Figure]

Specific comments:

1) Spelling and grammar should be improved in general. Sometimes the sentence construct is not concise enough. e.g. Page 1, line 15-17. 2) P2, Lines22, there were some references about the historical landscape changes about this research area, and please cite these literatures, for example, "Nian, Y. Y., X. Li, and J. Zhou. 2017. 'Landscape changes of the Ejin Delta in the Heihe River Basin in Northwest China from 1930 to 2010.' International Journal of Remote Sensing 38 (2): 537-57. doi: 10.1080/01431161.2016.1268732". 3) Page 2, line 38. Please add the newly published papers about the historical landscapes or cultivated area, especially in river basin scale. e.g. Xie et al., 2016. Assessing the evolution of oases in arid regions by reconstructing their historic spatio-temporal distribution: a case study of the Heihe River Basin, China. Frontiers of Earth Science; Hu and Li, 2014. Spatial distribution of an ancient agricultural oasis in Juyan, northwestern China. Frontiers of Earth Science, 8(3): 338-350. 4) P3, Line20, the moving and semi moving dunes should be revised as the words of the mobile dune and semi-mobile dune. 5) P3, Line21-22, when did these percent happen? please give more descriptions. 6) P4, Figures1, please mark the name of the meteorological stations. 7) P5, Line8, the Juyan Lake started to retain water again in 2002, please make sure this is correct or maybe you shall cite other reference about the lake restore. 8) P5, Line22, please use a table to list the data sources about Landsat iamges (path/row). 9) P5, Line25, pease delete the words of "during late summer and early autumn", and retain the words of "from June to October". 10) Yingluoxia Station is a key point to divide the upstream, midstream and downstream, but it is hard to identify the position of this station. Please change the symbol and color in the Figrue1. Furthermore, Heihe River Basin is your study area, so I would suggest to draw out the upstream area in figure 1 as well. 11) Page 5, line 17. The study area has a long history and many human activities took place during different periods. The ancient study periods given in this manuscript is incomplete. First, historical documents support evidence that there are human activities in the Heihe River Basin in Sui Dynasty, and this should be included. Also, you can combine those two dynasties in

Sui and Tang Dynasties. Second, there were prosperous human activities performed in the study area more than 150 years during Xixia Dynasty. I think the history of the Heihe River Basin should include the Xixia Dynasty. It is incomplete to ignore the above two important historical periods for HRB. Recommend to refer to some papers to get the information about the human activities developed in the downstream of the Heihe River Basin during the Xixia Dynasty. e.g., Hu and Li, 2014. Spatial distribution of ancient agricultural oasis in Juyan, northwestern China). 12) Page 6, line 9-14. In the method part of 2.3.1, you use the MODIS NDVI profile 2013 to calibrate the Landsat NDVIs to obtain annual maximum NDVI. Why do you chose the year of 2013, but not other years? 13) P 6, Figures 2, the equation in the figure should be separated from the figure. 14) Page 7, line 14-24. Please clarify when did you use the model established by Zhao to calculate the biomass based on NDVI. 15) P7, Lines20 and Line 21, Please list the reference "Zhao et al. 2006 and Zhao et al. 2010" in your references list. 16) Page 8-9, line 39-line 1. In the section of 3.1, the reconstructed natural and crop vegetation distribution after 1987 were interpreted from Landsat images. How to distinguish the natural and crop vegetation from Landsat images? And how about the accuracy? Please clarify the methods you used accordingly. 17) P10, from your results and Figure3, how to interpret (forcing mechanism) the vegetation change in the west river of the lower reaches of the Heihe River from Yuan Dynasty to Ming Dynasty, also how to interpret the vegetation change in the Gurina area from RC to 1987. 18) Page14, line 17-19. You divided the natural and crop vegetation development in the study area into three stages. Why the period of the Republic of China (RC) is included into the Pre-development stage? Because the results of the pre-development stage was conducted from the previous studies, while that of the Republic of China (RC) is interpreted from Landsat imageries. Meanwhile, I think there were some common features of the ancient periods from the Han Dynasty to the Qing Dynasty, exclusive the period of Republic of China (RC).So, so it could be better to category RC period as as a rapid development period. 19) I think the discussion will be more convincing if the authors can further discuss the relationships between the climate change and human

activities and the evolution of vegetation system. For example, how the climate influence the trend of vegetation changes can be analyzed according to the meteorological data or the previous studies. There is enough space to improve the discussion in the manuscript.

―――――――――――――――――――――

---

## Referee Comment (RC2) · Anonymous Referee #2 · 18 Apr 2017

The manuscript used Remote sensing data and historical research results to studied vegetation change in Heihe River basin in past 2000 years. The topic of the manuscript is very creative as there aren't many researches on historical vegetation change, especially on natural vegetation. The results of historical natural vegetation study is useful, however there are too many uncertainties to study. The authors' methods were not very complicated but proved to be very effective. The results of the manuscript were also very interesting. Considering the principle of HESS, I suggest accept the article after micro modification. 1. Page 3. Legends should be added to Figure 3. 2. Page 5. Did you considered other data source instead of Landsat images? Landsat images are not very useful to study interannual variability. In the article, MODIS data was used to rectify Landsat. It is acceptable, however, it also increased error of the results. 3. Page 7. Need more explanation for the data you quoted. 4. Page 14. Figure 7 is not

very clear to me. 5. In general view, natural vegetation and crops compete of water resources. Can you explain why in your results the natural vegetation and crops increase or decrease synchronously?

———————————————————

---

## Author Comment (AC1) · 15 May 2017

We would like to thank the reviewer for the positive feedback on our manuscript and we are grateful for the valuable critical comments which could further improve the manuscript. Listed below is our response to each comment. Full details of the revisions will be provided in the revised manuscript.

Referee #1's General comment This study aims to understand the evolution of vegetation system in Heihe River Basin over the past 2000 years, by reconstructing the spatiotemporal distribution of natural vegetation and crop vegetation from those previous study results and Landsat images. Because of the challenging problems in the human-environment interaction in this region, long-term change in vegetation system in the study area should be much appreciated. This research has made some advancement in the evolution of vegetation system with long-term scale, by synthesizing remote sensing data and results from previous researches. It will promote the understanding of the ecosystem health in the region and how to achieve a sustainable management in the Heihe River Basin. Overall, this is a solid and well documented contribution that can be accepted after some moderate revisions.

Response: We appreciate the reviewer's positive general comment on this study. We will revise the manuscript according to the reviewer's specific comments. The response to these comments is given as follows:

Referee #1's Specific comments 1) Spelling and grammar should be improved in general. Sometimes the sentence construct is not concise enough. e.g. Page 1, line 15-17.

Response: Thanks for the comment. We will go through the manuscript to improve the sentences where the construct is not concise enough.

In specific to the example provided (Page 1, line 15-17), we have revised the sentence as below: "The area and biomass of the vegetation before 1987 were reconstructed based on the relationships between vegetation and climatic and hydrological variables developed with instrumental data in the last 30 years."

In addition, after addressing all the comments, the manuscript will be edited by a native English-speaking expert to improve the English expressions of the whole manuscript.

2) P2, Lines22, there were some references about the historical landscape changes about this research area, and please cite these literatures, for example, "Nian, Y. Y., X. Li, and J. Zhou. 2017.'Landscape changes of the Ejin Delta in the Heihe River Basin in Northwest China from 1930 to 2010.' International Journal of Remote Sensing 38 (2): 537-57. doi: 10.1080/01431161.2016.1268732".

Response: Thanks for the comments and the suggested references. We will add these references to the list and include their major findings in the introduction section.

3) Page 2, line 38. Please add the newly published papers about the historical landscapes or cultivated area, especially in river basin scale. e.g. Xie et al., 2016. Assessing the evolution of oases in arid regions by reconstructing their historic spatio-temporal distribution: a case study of the Heihe River Basin, China. Frontiers of Earth Science; Hu and Li, 2014. Spatial distribution of an ancient agricultural oasis in Juyan, northwestern China. Frontiers of Earth Science, 8(3): 338-350.

Response: We agreed and will add these papers to the Reference section and include their major findings in the Introduction section.

4) P3, Line20, the moving and semi moving dunes should be revised as the words of the mobile dune and semi-mobile dune.

Response: We agreed. We will correct the "moving and semi moving dunes" to "mobile dune and semi-mobile dune" as suggested.

5) P3, Line21-22, when did these percent happen? please give more descriptions.

Response: Thanks for the comment. The percentage was calculated based on the land use and land cover data of the HRB in 2011. The LULC data were collected from the WestDC database. To clarify the point, we will include the necessary information in the revised manuscript.

6) P4, Figures1, please mark the name of the meteorological stations.

Response: We agreed. We will update Figure 1 with meteorological stations labelled with station names.

7) P5, Line8, the Juyan Lake started to retain water again in 2002, please make sure this is correct or maybe you shall cite other reference about the lake restore.

Response: Thanks for the comment. We checked published references and the Lake was reported to retain water again since 2002. We will cite references to confirm this point.

8) P5, Line22, please use a table to list the data sources about Landsat iamges (path/row).

Response: We agreed. We will provide a table in the revised manuscript listing the images used in the analysis. Information including imaging dates, data quality (e.g. Cloud coverage, signal noise) will be detailed in the table.

9) P5, Line25, please delete the words of "during late summer and early autumn", and retain the words of "from June to October".

Response: Agreed. We will revise the sentence accordingly.

10) Yingluoxia Station is a key point to divide the upstream, midstream and downstream, but it is hard to identify the position of this station. Please change the symbol and color in the Figrue1. Furthermore, Heihe River Basin is your study area, so I would suggest to draw out the upstream area in figure 1 as well.

Response: We agreed. We will further modify Figure 1 by adding upstream area of the Heihe River Basin and changing the symbol and color of Yingluoxia Station.

11) Page 5, line 17. The study area has a long history and many human activities took place during different periods. The ancient study periods given in this manuscript is incomplete. First, historical documents support evidence that there are human activities in the Heihe River Basin in Sui Dynasty, and this should be included. Also, you can combine those two dynasties in Sui and Tang Dynasties. Second, there were prosperous human activities performed in the study area more than 150 years during Xixia Dynasty. I think the history of the Heihe River Basin should include the Xixia Dynasty. It is incomplete to ignore the above two important historical periods for HRB. Recommend to refer to some papers to get the information about the human activities developed in the downstream of the Heihe River Basin during the Xixia Dynasty. e.g., Hu and Li, 2014. Spatial distribution of ancient agricultural oasis in Juyan, northwestern China).

Response: We agreed. The current six dynasties were selected according to the availability of vegetation maps for historic periods (Lu et al., 2015). We will include Sui by considering Sui and Tang as a single period as Sui-Tang. The situation was similar for the Xixia Dynasty and it could be combined into Yuan Dynasty..We will refer to the datasets and add the data for Sui and Xixia into the analysis in the revised manuscript.

12) Page 6, line 9-14. In the method part of 2.3.1, you use the MODIS NDVI profile 2013 to calibrate the Landsat NDVIs to obtain annual maximum NDVI. Why do you chose the year of 2013, but not other years?

Response: The reason why the year of 2013 was chosen is because 2013 is the final year we have Landsat images and it was naturally selected for this analysis. In addition, we made an assumption that vegetation growth would follow a similar phenology cycle in each year. Thus, data from different years would not cause significant differences in final results.

13) P 6, Figures 2, the equation in the figure should be separated from the figure.

Response: Agreed, the equation in the figure will be separated from the figure.

14) Page 7, line 14-24. Please clarify when did you use the model established by Zhao to calculate the biomass based on NDVI.

Response: We will clarify the period when we used the model established by Zhao to calculate the biomass based on NDVI and add details on how we calculated natural vegetation biomass and crops biomass of the study area

15) P7, Lines20 and Line 21, Please list the reference "Zhao et al. 2006 and Zhao et al. 2010" in your references list.

Response: Thanks. We will add the reference to the reference list.

16) Page 8-9, line 39-line 1. In the section of 3.1, the reconstructed natural and crop vegetation distribution after 1987 were interpreted from Landsat images. How to distinguish the natural and crop vegetation from Landsat images? And how about the accuracy? Please clarify the methods you used accordingly.

Response: The methods we used to distinguish natural and crop vegetation were provided in section 2.3.2. In brief, two rounds of threshold analysis were applied: the first threshold was used to mask out non-vegetation pixels; the second threshold was used to separate natural vegetation from crop vegetation. The threshold values were chose mainly based on analyzing the NDVI histogram distribution characteristics. The derived results were first compared to the corresponding Landsat images to check their accuracy in each year. In addition, the results in 2000 and 2011 were verified with existing land use maps (2000, 2011) obtained from the WestDC database. The calculated kappa coefficients (k) were 0.7206 and 0.6731 for 2000 and 2011, respectively.

17) P10, from your results and Figure3, how to interpret (forcing mechanism) the vegetation change in the west river of the lower reaches of the Heihe River from Yuan Dynasty to Ming Dynasty, also how to interpret the vegetation change in the Gurina area from RC to 1987.

Response: Thanks for the valuable comments. Among the reconstructed climatic and hydrological records in the study area, the variables which presented substantial changes were the annual average temperature and streamflow. The temperature kept decreasing from the early Yuan Dynasty to the end of the Ming Dynasty (Tang et al., 2017) while streamflow kept increasing (Sakai et al, 2012). Therefore, more streamflow and reduced evaporation possibly made more water available for local vegetation growth for this period. This could be the reason for the expansion of natural vegetation along the West River.

The vegetation change in the Gurina Area could be the results of different methods applied for previous dynasties and for recent years. For the reconstructions studies in historical periods, the effort was concentrated along the river channels while vegetation in remote areas was much less emphasized. Whereas in recent years with remotely

sensed data used, vegetation in remote area could be identified much more easily and with higher precision. We will add this analysis in Discussion Section as the limitations of our methods in the revised manuscript.

18) Page14, line 17-19. You divided the natural and crop vegetation development in the study area into three stages. Why the period of the Republic of China (RC) is included into the Pre-development stage? Because the results of the pre-development stage was conducted from the previous studies, while that of the Republic of China (RC) is interpreted from Landsat imageries. Meanwhile, I think there were some common features of the ancient periods from the Han Dynasty to the Qing Dynasty, exclusive the period of Republic of China (RC).So, so it could be better to category RC period as as a rapid development period.

Response: Thanks for the comment. As revealed with the trend of natural and crop vegetation areas, RC could be recognized either as the end of a period with low coverage, or as the start of a period with rapid vegetation expansions. We agree when considering the common features of RC, it is more appropriate to categorize it as a rapid development period. We will revise the manuscript accordingly.

19) I think the discussion will be more convincing if the authors can further discuss the relationships between the climate change and human activities and the evolution of vegetation system. For example, how the climate influence the trend of vegetation changes can be analyzed according to the meteorological data or the previous studies. There is enough space to improve the discussion in the manuscript.

Response: Thanks for the comments. We will improve the discussion section in the revised manuscript by analyzing the relationships between the climate change and human activities and the evolution of vegetation system, adding the limitations of our research and implications for future research.

---

## Author Comment (AC2) · 15 May 2017

Referee #2's General comments The manuscript used Remote sensing data and historical research results to studied vegetation change in Heihe River basin in past 2000 years. The topic of the manuscript is very creative as there aren't many researches on historical vegetation change, especially on natural vegetation. The results of historical natural vegetation study is useful, however there are too many uncertainties to study. The authors' methods were not very complicated but proved to be very effective. The results of the manuscript were also very interesting.

Response: Thanks for the reviewer' positive comments on our study.

Referee #2's Specific comments:

1) Page 3. Legends should be added to Figure 3.

Response: Agreed. We will update Figure 3 with legends in the revised manuscript.

2) Page 5. Did you considered other data source instead of Landsat images? Landsat images are not very useful to study interannual variability. In the article, MODIS data was used to rectify Landsat. It is acceptable, however, it also increased error of the results.

Response: Thanks for the comment. Although there are other satellite images available such as MODIS and AVHRR, we would like to argue that Landsat image is one of the most suitable sources for long term studies as it has the longest archived data available and with relative stable qualities. In addition, we set criteria in selecting the images for example it should be acquired in the same period of each year (growing season) and there was low cloud contamination during the acquisition period. In rectifying Landsat with MODIS, it might result in certain uncertainties since there are differences in the amplitudes of Landsat and MODIS signals captured at different spatial scales. However, the phenology cycles of vegetation obtained by the two datasets should be similar. Therefore, we determined the thresholds using the rectified images rather than the original ones acquired on different dates.

3) Page 7. Need more explanation for the data you quoted. Response: Agreed. We will revise this section to clearly demonstrate the data we quoted and the data we reconstructed, respectively. Specifically, we will list the data we extracted from previous studies (e.g. historical vegetation distributions, stream flow and precipitations etc.) and give necessary introductions and citations. We will also give comprehensive descriptions about the reconstructed datasets (biomass) by detailing the hypothesis and calculations.

4) Page 14. Figure 7 is not very clear to me. Response: Thanks. We will include more details in the caption to clearly provide information about the scatter diagram. We will increase the font size and resolution of the figure in the revised manuscript as well.

[Figure]

5) In general view, natural vegetation and crops compete of water resources. Can you explain why in your results the natural vegetation and crops increase or decrease synchronously?

Response: Thanks for the comment. We agreed that in general view natural vegetation and crops compete of water resources when the amount of total water resources is fixed. However when water resources available are increased, the competition could be weakened or even disappear for a certain period. For example, in our study area, for the recent decades, increasing streamflow caused by the elevated temperature and increased snow melting in the upstream has caused the increase of total water resources which might be the reason for the overall growth of both natural and crop vegetation. However, the competition between natural vegetation and crop was observed as indicated in Figure 4. For instance, crop area for the period from Yuan Dynasty to RC showed constant increase, whereas natural vegetation distributions decreased for the same period. In 1998 - 2003 decrease in natural vegetation were observed as well. As we argued in the Discussion section, further research should be conducted to clarify this phenomenon for the mechanistic perspective.

---

## Author Comment (AC3) · 15 May 2017

Your paper focuses mostly on vegetation change. You have submitted this paper to a journal dealing with water resources in general, and hydrological issues in particular. You will need to demonstrate the relevance of your paper to the improved understanding of hydrological systems or water resources.

Response: Thanks for this valuable comment for us to sharpen key points of our manuscript. We think that our manuscript has made important contribution to the improvement of understanding of hydrological systems/water resources. Firstly, water-vegetation relationships are the most important hydrological issue in the desert hydrological systems where oasis is a unique landscape. The history of cultivated oasis development is the process in which humans placed stress on the natural hydrological system. Secondly, as Reviewer 2 commented that "the topic of the manuscript is very creative as there aren't many researches on historical vegetation change, especially on natural vegetation". Our manuscript has provided an understanding of how the vegetation system co-evolved with the changes of hydrological system for a time frame of 2000 years, which is of great implication for future water resources management. Finally, our findings have clarified a common sense that natural vegetation and crops compete on water resources in arid regions. Our results showed that increasing streamflow caused by the elevated temperature and increased snow melting in the upstream may lead to the overall growth of both natural and crop vegetation during the same period. However, we do agree with the Editor that we should reshape our manuscript to highlight our contribution to the understanding of hydrological system in our revised manuscript.

---

## Author Response (AR1)

As the editor and the referees all mentioned the spelling and grammar in our manuscript, a native English-speaking expert have edited it to improve the English expressions of the whole manuscript. Follows are some specific changes based on the editor and referees' comments.

Edit based on the editor's comments:

P1L14-17: Changed the sentence to "Archived Landsat images, historical land use maps and hydrological records were introduced to derive the long term spatial distribution of natural and crop vegetation and the corresponding biomass levels."

P1L22-24: Changed the last sentence of Abstract to "This ratio reflects the reaction of land and water development to a changing climate and an altering social-economic conditions at the river basin level, therefore, it could be used as an indicator for water and land management at river basins."

P2L10: Changed "water conditions" to "water availability conditions"

P2L14-16: Changed the sentence to "However, few studies have investigated how the vegetation system evolved in response to the changes in water regimes at the basin scale."

P2L23: Changed "missed" to "lacking"

P2L28: Changed "provides" to "provide"

P2L32: Changed "to reflect the status of the vegetation and widely" to "which have been widely"

P2L35: Changed "enabled the possibilities of reconstructions of" to "been applied to reconstruct"

P3L2: Changed "the way of" to "use of"

P3L18: Changed "of" to "in"

P3L21: Changed "such as Gobi desert" to "such as the desert, Gobi..."

P3L22: Changed "is" to "are"

P4L8: Changed "could" to "can"

P4L9: Changed "fetch up" to "reach up"

P4L13: Changed the sentence to "River inflow from the North Qilian Mountain constituted the primary water source for the river basin."

P5L3: Changed "elevated" to "increasing"

P7L20-21: Added the units of biomass to the equations

P7L32: Edited the paragraph to "The crop and natural vegetation areas in previous dynasties (Table 1) were derived from Lu (2015)'s results. In Lu's study, historical cropland was reconstructed based on population, grain yield and ancient ruin distributions (Lu *et al.*, 2015). Natural vegetation distributions were estimated based on the assumption that people tended to select natural oases (grassland and forest) rather than desert for reclamation in historical periods because the former have better water and soil conditions in these arid regions, while the abandoned croplands desertified. Thus, natural vegetation for previous dynasties could be evaluated based on the changes in cropland between the previous and current dynasties (Lu *et al.*, 2015)."

Edit based on referees' comments:

Referee 1

2) P2, Lines22, there were some references about the historical landscape changes about this research area, and please cite these literatures, for example, "Nian, Y. Y., X. Li, and J. Zhou. 2017.' Landscape changes of the Ejin Delta in the Heihe River Basin in Northwest China from 1930 to 2010.' International Journal of Remote Sensing 38 (2): 537-57. doi:

10.1080/01431161.2016.1268732".

We add "To a centurial scale, Nian (2017) found that the rapid expansion of cultivated land was the primary force causing serious ecological deterioration in the HRB." to the Introduction Section.

4)    P3, Line20, the moving and semi moving dunes should be revised as the words of the mobile dune and semi-mobile dune.

We correct the "moving and semi moving dunes" to "mobile dunes and semi-mobile dunes" as suggested.

5)    P3, Line21-22, when did these percent happen? please give more descriptions.

We add "according to land cover map of 2011, data available at WestDC database http://westdc.westgis.ac.cn/" to the sentence.

6)    P4, Figures1, please mark the name of the meteorological stations.
10)  Yingluoxia Station is a key point to divide the upstream, midstream and downstream, but it is hard to identify the position of this station. Please change the symbol and color in the Figrue1. Furthermore, Heihe River Basin is your study area, so I would suggest to draw out the upstream area in figure 1 as well.

We updated Figure 1 with: adding meteorological stations labelled with station names; adding upstream area of the Heihe River Basin; changing the symbol and color of Yingluoxia Station.

7)    P5, Line8, the Juyan Lake started to retain water again in 2002, please make sure this is correct or maybe you shall cite other reference about the lake restore.

We added the references "(Nian *et al.*, 2017)".

9)    P5, Line25, please delete the words of "during late summer and early autumn", and retain the words of "from June to October".

We changed "during late summer and early autumn" to "from June to October"

11)  Page 5, line 17. The study area has a long history and many human activities took place during different periods. The ancient study periods given in this manuscript is incomplete. First, historical documents support evidence that there are human activities in the Heihe River Basin in Sui Dynasty, and this should be included. Also, you can combine those two dynasties in Sui and Tang Dynasties. Second, there were prosperous human activities performed in the study area more than 150 years during Xixia Dynasty. I think the history of the Heihe River Basin should include the Xixia Dynasty. It is incomplete to ignore the above two important historical periods for HRB. Recommend to refer to some papers to get the information about the human activities developed in the downstream of the Heihe River Basin during the Xixia Dynasty. e.g., Hu and Li, 2014. Spatial distribution of ancient agricultural oasis in Juyan, northwestern China).

We edited the "Tang dynasty" to "Sui-tang Dynasty" in the manuscript. And at the end of our manuscript, we add "Xixia Dynasty ruled the area for more than 150 years (AD 1038 - 1227) and prosperous human activities were recorded which might cause substantial changes to both crop and natural vegetation. Existing literature reported crop distribution in the lower reaches of HRB in Xixia using archaeological methods (Hu and Li, 2014), but data for crop and natural vegetation covering the entire basin is lacking in both literature and historical documents." to describe our limitation in Xixia dynasty.

13) P 6, Figures 2, the equation in the figure should be separated from the figure.

The equation in the figure was separated from the figure.

17) P10, from your results and Figure3, how to interpret (forcing mechanism) the vegetation change in the west river of the lower reaches of the Heihe River from Yuan Dynasty to Ming Dynasty, also how to interpret the vegetation change in the Gurina area from RC to 1987.

We add "The consistently increasing streamflow due to the warm climate (more notably for the past two decades) and therefore increased precipitation and snow melting in upper reaches might have supported the expansion of crop vegetation. The increased streamflow might have supported the rapid natural vegetation growth in the middle reaches as well, either through direct watering of the river side vegetation system, or through water leakage from crop irrigation areas. However, vegetation in the lower reaches did not show synchronous development. Obviously, overuse of water in the middle reaches was the primary contributor which coheres with the existing literature (Nian *et al.*, 2017; Zhao *et al.*, 2016; Cheng 2002; Wang *et al.*, 2007)." to the Discussion Section.

19) I think the discussion will be more convincing if the authors can further discuss the relationships between the climate change and human activities and the evolution of vegetation system. For example, how the climate influence the trend of vegetation changes can be analyzed according to the meteorological data or the previous studies. There is enough space to improve the discussion in the manuscript.

We add "Since 2000 (the post-development stage), crop vegetation distribution has slightly increased but natural vegetation has experienced a relatively faster increase. This could be attributed to two major reasons. The first is the elevated temperature and increased streamflow provided sufficient water for both crop and natural vegetation development, this could be evidenced by their significant relationship with both natural and crop vegetation areas (Figure7 c, d). The second is owing to the implementation of the water reallocation policy which aimed to "secure water supply to the lower course of the basin to avoid ecosystem degradation"." to the Discussion Section.

Referee 2
1) Page 3. Legends should be added to Figure 3.

We updated Figure 3 with legends.

3    2)  Page 7. Need more explanation for the data you quoted.

We rewrite P7L26-34 to "The crop and natural vegetation areas in previous dynasties (Table 1) were
derived from Lu (2015)'s results. In Lu's study, historical cropland was reconstructed based on
population, grain yield and ancient ruin distributions (Lu *et al.*, 2015). Natural vegetation
distributions were estimated based on the assumption that people tended to select natural oases
(grassland and forest) rather than desert for reclamation in historical periods because the former
have better water and soil conditions in these arid regions, while the abandoned croplands desertified.
Thus, natural vegetation for previous dynasties could be evaluated based on the changes in cropland
between the previous and current dynasties (Lu *et al.*, 2015).".

3)  Page 14. Figure 7 is not very clear to me.
We included more details in the caption to clearly provide information about the scatter
diagram and increased the font size and resolution of the figure.

4)  In general view, natural vegetation and crops compete of water resources. Can you explain why
in your results the natural vegetation and crops increase or decrease synchronously?

[revised manuscript text omitted]